# Characterization of Metabolic Correlations of Ursodeoxycholic Acid with Other Bile Acid Species through In Vitro Sequential Metabolism and Isomer-Focused Identification

**DOI:** 10.3390/molecules28124801

**Published:** 2023-06-16

**Authors:** Wei Li, Wei Chen, Xiaoya Niu, Chen Zhao, Pengfei Tu, Jun Li, Wenjing Liu, Yuelin Song

**Affiliations:** 1Modern Research Center for Traditional Chinese Medicine, Beijing Research Institute of Chinese Medicine, Beijing University of Chinese Medicine, Beijing 100029, China; lw160221071@163.com (W.L.);; 2Zhangzhou Pien Tze Huang Pharmaceutical Co., Ltd., Zhangzhou 363000, China; 3School of Pharmacy, Henan University of Chinese Medicine, Zhengzhou 450046, China

**Keywords:** ursodeoxycholic acid, metabolites, isomeric identification, conjugation site, squared energy-resolved mass spectrometry

## Abstract

As a first-line agent for cholestasis treatment in a clinic, ursodeoxycholic acid rectifies the perturbed bile acids (BAs) submetabolome in a holistic manner. Considering the endogenous distribution of ursodeoxycholic acid and extensive occurrences of isomeric metabolites, it is challenging to point out whether a given bile acid species is impacted by ursodeoxycholic acid in a direct or indirect manner, thus hindering the therapeutic mechanism clarification. Here, an in-depth exploration of the metabolism pattern of ursodeoxycholic acid was attempted. Sequential metabolism in vitro with enzyme-enriched liver microsomes was implemented to simulate the step-wise metabolism and to capture the metabolically labile intermediates in the absence of endogenous BAs. Squared energy-resolved mass spectrometry (ER^2^-MS) was utilized to achieve isomeric identification of the conjugated metabolites. As a result, **20** metabolites (**M1**–**M20**) in total were observed and confirmatively identified. Of those, eight metabolites were generated by hydroxylation, oxidation, and epimerization, which were further metabolized to nine glucuronides and three sulfates by uridine diphosphate-glycosyltransferases and sulfotransferases, respectively. Regarding a given phase II metabolite, the conjugation sites were correlated with first-generation breakdown graphs corresponding to the linkage fission mediated by collision-induced dissociation, and the structural nuclei were identified by matching second-generation breakdown graphs with the known structures. Together, except for intestinal-bacteria-involved biotransformation, the current study characterized BA species directly influenced by ursodeoxycholic acid administration. Moreover, sequential metabolism in vitro should be a meaningful way of characterizing the metabolic pathways of endogenous substances, and squared energy-resolved mass spectrometry is a legitimate tool for structurally identifying phase II metabolites.

## 1. Introduction

Bile acids (BAs) are derivatives of hepatic cholesterol metabolites, which play a key role in nutrient absorption and the biliary secretion of lipids, toxic metabolites, and xenobiotics [1,2,3,4]. They also serve as significant regulators in mediating numerous physiological and pathophysiological processes through activating some nuclear receptors, such as farnesoid X receptor (FXR), pregnane X receptor (PXR), and Takeda G-protein-coupled receptor 5 (TGR5). These nuclear receptors function in the transcriptional regulation of genes involved in bile acid synthesis, transport, and metabolism [5,6,7]. The diversity and composition of the bile acid pool are related to their roles in diseases. Primary BAs are synthesized by the liver via a series of enzymatic reactions from cholesterol and then released into the small intestine. Furthermore, the gut microbiota is also reported to enrich the diversity of the second BAs structures through four pathways: deconjugation of glycine or taurine, dehydrogenation, dehydroxylation, and epimerization [8,9,10,11]. Recently, some novel amino-acid-conjugated BAs, other than nor-DCA, tetra-BAs, 3,7,12,24-tetrol-26,27-dinorcholetane-24-sulfate, have been mined in gut microbiota metabolism, which further enrich the diversity and composition of the BA pool [12,13,14]. Regarding the free BAs, oxidation frequently occurs at C-3, C-6, C-7, and C-12 sites. For conjugated BAs, tauro and glycol substitutions always occur for the C-24 carboxy group. However, the identification of BAs isomers remains a challenge.

Ursodeoxycholic acid (UDCA) is a free type of bile acid, which has been utilized for medical purposes for many years [15]. While it is considered a secondary bile acid in humans, UDCA is a primary acid derived from the host in other animal species, such as bears and mice [16]. It has been widely applied in clinical treatment for various liver disorders, such as primary biliary cholangitis (PBC) [17] and intrahepatic cholestasis [18], as well as others. Very recently, a study has shown that UDCA may have pharmacological effects against COVID-19 by reducing ACE2 levels and SARS-CoV-2 infection [19]. Nevertheless, the effects of UDCA on the enterohepatic circulation of BAs have yet to be fully investigated. The bile acid pool plays a critical role in reflecting the physiological or pathological status of an organism, especially the liver, together with intestine organs where most bile acid metabolism occurs. To address this gap, we conducted a sequential metabolism of UDCA in vitro and clarified how these metabolic processes occur in vivo with other BAs. This research may help us better understand the effects of UDCA administration on the enterohepatic circulation of BAs and provide insight into its potential therapeutic application.

With the rapid advancement of the MS/MS technique, it is currently favored as a cutting-edge tool for metabolites’ characterization owing to its superior sensitivity, as well as the unique advantage of generating rich structural information, primarily in terms of high-resolution (HR) mass-to-charge ratio (*m*/*z*). The online energy-resolved MS (online ER-MS) could provide the molecular descriptors, such as CE_50_ values and optimal collision energy (OCE), to determine the conjugated sites [20]. Recently, multiple-dimension MS analysis has become more and more popular, particularly in natural product structure identification at home and abroad. For instance, Qtrap-MS is a hybrid mass spectrometer, which combines the features of triple-quadrupole (QQQ) and ion trap (IT) instruments. It is composed of a quadrupole cell and a linear ion trap (LIT) chamber, which offers superior sensitivity and selectivity for targeted metabolomics analysis. Undoubtedly, it serves as the best choice for performing two consecutive online ER-MS assays, known as ER^2^-MS, which provide more detailed substructural information about the metabolites being analyzed. Compared with multiple-reaction monitoring (MRM) corresponding to the first ER-MS measurement [21], the MRM cubed (MRM^3^) algorithm exhibits suitable characteristics for meeting the second ER-MS assay requirement. The MRM^3^ algorithm allows the selection of specific fragmentation patterns and detection of diagnostic ions based on their unique structural characteristics. This approach improves the specificity and selectivity of the analysis and enables the identification of metabolites with greater accuracy. For instance, for BA glucuronides, the glucuronidation position can be identified by applying the first online ER-MS program to [M−H]^−^ > [M−H−C_6_H_8_O_6_]^−^ because a molecular descriptor, namely OCE, is tightly correlated with the glucuronidation site. Subsequently, [M−H]^−^ > [M−H−C_6_H_8_O_6_]^−^ > [M−H−C_6_H_8_O_6_]^−^ participates in the MRM^3^ assay, and the second ER-MS measurement is accomplished by assigning progressive AF2 values to a panel of comparable MS^3^ experiments. As a result, BA glucuronides could be identified through ER^2^-MS.

Here, we conducted an experiment to investigate the sequential metabolism of UDCA in vitro and identify its transformation metabolites using liver microsomes from humans and mice, which are enriched with fruitful phase I and II metabolic enzymes. UDCA-*d*_4_ was performed to validate the metabolic products of UDCA. Subsequently, the confidential metabolic structures were identified through ER^2^-MS, including OCE and AF2 values molecular descriptors. To detect trace metabolites, the routine analytical pipeline Qtrap-MS with *p*MRM acquisition mode was performed subsequently and captured the diagnostic fragmentation ions. Herein, an integrated workflow (Figure 1) was proposed to eliminate the process of the experiment. Additionally, Q Exactive-MS was utilized to acquire the high-resolution MS/MS information. The findings acquired in this study will assist in enhancing our comprehension of the biotransformation bonds associated with other bile acids, not merely UDCA. Furthermore, ER^2^-MS enabled us to obtain an analytical pathway to enhance the potential for identifying isomers via LC−MS.

## 2. Results and Discussion

### 2.1. Identification of In Vitro Sequential Metabolites of UDCA

In vitro sequential metabolism of UDCA could provide an insight into the transformation of bile acids into each other, particularly in a case where UDCA was administered. In some way, this process may also occur naturally and the real condition could be performed in vivo. In total, **20** metabolites were tentatively identified by high-resolution MS combined with the ER^2^-MS strategy (Appendix A).

**LC–MS/MS behaviors of BAs.** The mass fragmentation rules were clearly summarized in the literature [22,23,24] as follows. Unconjugated BAs were derived from cholic acid (CA) or CDCA via various biosynthetic pathways. Free BAs mainly participated in phase I metabolism of UDCA, which included oxidation, epimerization, and hydroxylation. It is always observed that dehydroxylation and decarboxylation on the sterol backbone result in neutral losses of H_2_O (18 Da), CO (28 Da), and CO_2_ (44 Da). Conjugated BAs are formed through the glycine and taurine modification of the terminal carboxyl group of free BAs. Additionally, the sulfated and glucuronidated groups occur in the hydroxyl group sites of BAs corresponding to characteristic neutral losses of SO_3_ (80.06 Da) and glucuronic acid (176.12 Da). Furthermore, our group established an in-house database containing 201 BAs, which was subsequently employed to characterize metabolites according to all the aforementioned rules.

**Structural characterization of UDCA metabolites using LC–MS/MS information.** For UDCA phase I metabolites, there were eight metabolites identified as **M1**–**M8** (Appendix A), namely 7*β*-hydroxy-3-oxo-5*β*-cholan-24-oic acid (**M1**), 7-ketolithocholic acid (**M2**), CDCA (**M3**), 3*β*,7*α*-dihydroxy-6-oxo-5*β*-cholan-24-oic acid or 3*β*,6*α*-dihydroxy-7-oxo-5*β*-cholan-24-oic acid (**M4**), 3*α*,7*β*-dihydroxy-12-oxo-5*β*-cholan-24-oic acid (**M5**), 3*β*,7*β*,12*α*-trihydroxy-5*β*-cholan-24-oic acid (**M6**), ursocholic acid (**M7**), and *β*-MCA (**M8**). After the NADPH regeneration system, all samples were concentrated to dryness with nitrogen gas. Then, the residues of phase I products served as phase II substrates to carry out the reaction. For UDCA phase II metabolites, there were only three metabolites identified as **M13**, **M15**, and **M19,** including UDCA-3-*O*-glucuronide (**M13**), *β*-MCA-3-*O*-glucuronide (**M15**), and UDCA-3-sulfate (**M19**) (Appendix A).

**M1** and **M2** were taken as cases to clarify the structural annotation of metabolites, which shared an identical [M − H]^−^ ion at *m*/*z* 389.2702, and the elemental composition was calculated as C_24_H_38_O_4_. In comparison to prototype UDCA, oxidation should be responsible for reducing two hydrogen atoms. When Q1 ions entered the collision chamber to perform collision-induced dissociation, the primary fragment ions were observed as *m*/*z* 371.2637 (C_24_H_35_O_3_^−^), *m*/*z* 343.2656 (C_23_H_35_O_2_^−^), *m*/*z* 325.2553 (C_23_H_33_O^−^), and *m*/*z* 69.0346 (C_4_H_6_O^−^). These ions mainly orientated from neutral cleavages of H_2_O, HCOOH, and ring-A from C1-C10 to C4-C5. Among them, only **M1** occurred with *m*/*z* 69.0346, which indicated the dehydrogenation of the C3-OH group. Therefore, **M1** was identified as the C3-OH oxidation of UDCA, forming 7*β*-hydroxy-3-oxo-5*β*-cholan-24-oic acid. This conclusion was validated by the spectrum of UDCA-*d*_4_, in which the cleavage of ring-A from C1-C10 to C4-C5 was detected. Undoubtedly, **M2** was identified as the C7-OH group dehydrogenated metabolite, attributed to 7-ketolithocholic acid. The mass fragmentation pathways of **M1** and **M2** are proposed in Appendix A. Additionally, **M7** was identified as UCA compared to the authentic compound with retention time.

**BAs mining based on *p*MRM strategy.** In order to dig and clarify in depth the UDCA in vitro metabolism profiles involved in phase II reactions, the *p*MRM scan mode was carefully developed to identify the comprehensive metabolites, especially for trace metabolic products. As a result, a total of twenty metabolites were detected and tentatively characterized as **M1**–**M20** (Appendix A) in the negative ion mode, while without UDCA being involved, HLM (Appendix A) or MLM (Appendix A) was detected with the NADPH incubation system in vitro. Additionally, glucuronidation (Appendix A) or sulfation (Appendix A) assays were performed without UDCA involvement in HLM or MLM detection with the UDPGA or PAPS incubation system. For the phase I reaction, the **M1**–**M8** metabolites are shown in Figure 2A, of which the isotope labeling metabolites were spotted correspondingly via UDCA-*d*_4_ in vitro incubation (Appendix A). In terms of phase II metabolism containing glucuronidation and sulfonation, the compounds comprising carboxy groups or phenolic hydroxyl groups tend to combine with glucuronic acid or sulfate. Hence, there were nine glucuronide products shown in Figure 2B (**M9**–**M17**), including 7*β*-hydroxy-3-oxo-5*β*-cholan-24-oic acid-7-*O*-glucuronide (**M9**), 7*β*-hydroxy-3-oxo-5*β*-cholan-24-oic acid-24-*O*-glucuronide (**M10**), 7-ketolithocholic acid-24-*O*-glucuronide (**M11**), UDCA-7-*O*-glucuronide (**M12**), UDCA-3-*O*-glucuronide (**M13**), UCA-7-*O*-glucuronide (**M14**), *β*-MCA-3-*O*-glucuronide (**M15**), UCA-3-*O*-glucuronide (**M16**), 3*β*,7*β*,12*α*-trihydroxy-5*β*-cholan-24-oic acid-3-*O*-glucuronide (**M17**). Then, three sulfated metabolites were observed (Figure 2B)—a panel of bile acids whose hydroxy groups were modified through the sulfuric acid—yielded from the NADPH reaction system, including 7*β*-hydroxy-3-oxo-5*β*-cholan-24-oic acid-7-sulfate (**M18**), UDCA-3-sulfate (**M19**), and UDCA-7-sulfate (**M20**). Among them, **M2**, **M3**, **M7**, and **M8** were confidentially identified by authentic compounds with retention time and mass fragment information. However, a group of isomers, **M14** (*t*_R_ = 7.78 min), **M15** (*t*_R_ = 8.44 min), **M16** (*t*_R_ = 8.62 min), and **M17** (*t*_R_ = 9.2 min), were captured by *m*/*z* 583.31 > 407.28 with the *p*MRM program (Figure 3A). Only **M15** was hunted by HR-MS and observed primary MS^1^ at *m*/*z* 583.3144 ([M–H]^−^) and fragment ions, including 407.2813, 175.0255, and 113.0250 (Figure 3B), corresponding to C_24_H_39_O_5_^−^, C_6_H_7_O_6_^−^, and C_5_H_5_O_3_^−^. Therefore, **M15** was tentatively characterized as the 3*β*,7*β*,12*α*-trihydroxy-5*β*-cholan-24-oic acid-glucuronide, UCA-glucuronide, or *β*-MCA-glucuronide combined with those well-defined mass cracking rules and phase I metabolism. **M12** and **M13** were identified as UDCA-glucuronide or CDCA-glucuronide. Similarly, **M9**, **M10**, and **M11** were identified as 7*β*-hydroxy-3-oxo-5*β*-cholan-24-oic acid-glucuronide or 7-ketolithocholic acid-glucuronide. **M18** and **M20** were determined as 7*β*-hydroxy-3-oxo-5*β*-cholan-24-oic acid-sulfate and UDCA-sulfate. **M19** was characterized as UDCA-3-sulfate through the incubation of UDCA with human recombinase SULT2A1 in our laboratory. For phase II metabolites, the isotope labeling metabolites were also detected correspondingly via UDCA-*d*_4_ in vitro incubation (Appendix A), except for **M19**. In order to solve the characterization of isomeric metabolites, we further established a strategy based on ER^2^-MS for in-depth confirmation of the sites of those conjugated metabolites.

### 2.2. Phase II Metabolites’ Identification by ER^2^-MS

Although the MS analysis provided the information on the free and conjugated type BAs, the accurate characterization of UDCA phase II metabolites containing isomers remains challenging. In our group, extensive efforts were made to strengthen the structural annotation evidence via ER^2^-MS measurements, which were established to achieve as many as 201 BAs [25]. Herein, we proposed an *p*MRM interpretation strategy in an ER^2^-MS manner, which mainly aims to synergistically screen the conjugated sites and substructures of all phase II metabolites.

**Identity metabolites’ consolidation via application of ER^2^-MS.** The first ER-MS strategy was used to assign the glucuronide-conjugated site and the sulfate-conjugated site. Second-generation breakdown graphs were employed to confirm the scaffold, which was featured by the configuration and the location of carbonyl and/or hydroxy groups. **M15** was employed as a representative case. After undergoing the first ER-MS assay, OCE corresponding to *m*/*z* 583.3 > 407.3 was calculated to be −51.33 eV from a Gaussian-shaped first-generation breakdown graph (Figure 3C), located exactly within the OCE window 3-G rather than the ranges derived from 7-G and 24-G. The OCE distribution diagram was rendered through assaying authentic compounds in our group [25]. Consequently, glucuronide occurred for C3-OH. Following the second ER-MS assay, a sigmoid-shaped second-generation breakdown graph resulting from *m*/*z* 583.3 > 407.3 > 407.3 matched well with the one generated by applying *m*/*z* 407.3 > 407.3 > 407.3 to *β*-MCA (**M8**), one of the UDCA phase I metabolites (Figure 3D). However, **M14**, **M16**, and **M17** were captured as an isomer set through predefined ion transition by 583.3 > 407.3 simultaneously. The OCE corresponding to *m*/*z* 583.3 > 407.3 for **M15** isomers were calculated to be −50.57 eV, −53.74 eV, and −54.11 eV, respectively, from the first breakdown graph, located exactly within the OCE window 3-G. According to the UDCA phase I metabolites for *m*/*z* 407.3 > 407.3, there were only two hydroxyl sites conjugated with glucuronide, except for **M15**. Following the second ER-MS assay, a sigmoid-shaped second-generation breakdown graph resulting from *m*/*z* 583.3 > 407.3 > 407.3 to **M14**, **M16**, and **M17** matched well with the phase I metabolites generated by applying *m*/*z* 407.3 > 407.3 > 407.3 to **M7** and **M6**. As a result, **M14**, **M16,** and **M17** were tentatively identified as UCA-7-*O*-glucuronide, UCA-3-*O*-glucuronide, and 3*β*,7*β*,12*α*-trihydroxy-5*β*-cholan-24-oic acid-3-*O*-glucuronide.

**M9**, **M10,** and **M11** were captured as an isomeric set through predefined ion transitions as *m*/*z* 565.3 > 389.3 at 9.16 min, 9.37 min, and 9.52 min. They were predefined as the glucuronide-conjugated metabolites from **M1** or **M2** after phase I bioconversion. Then, great efforts were made to confirm the glucuronide-conjugated site, namely C3-OH, C7-OH, or C24-COOH. The following OCEs of the diagnostic ion at *m*/*z* 389.3 were calculated to be −47.68 eV, −48.38 eV, and −49.74 eV for the first breakdown graph. Then, we observed the OCEs of these isomers located exactly within the OCE window 7-G and 24-G (Figure 4A). In depth, the scaffolds of **M9**, **M10**, and **M11** were confirmatively configured as **M1** and **M2** (Figure 4B). Therefore, the structures of **M9**, **M10**, and **M11** were respectively identified as 7*β*-hydroxy-3-oxo-5*β*-cholan-24-oic acid-7-*O*-glucuronide, 7*β*-hydroxy-3-oxo-5*β*-cholan-24-oic acid-24-*O*-glucuronide, and 7-ketolithocholic acid-24-*O*-glucuronide.

**M19** and **M20** were captured through predefined ion transitions as *m*/*z* 469.3 > 97 and at 8.69 min and 8.92 min. For one of the metabolites (**M19)**, its primary mass fragment ion species of [M–H]^−^ (*m*/*z* 567.3202) was observed at *m*/*z* 471.2427, 96.9605, and 79.9576 (Appendix A). After converting the *m*/*z* values to elemental compositions, **M19** and **M20** were regarded as the sulfated products of UDCA or CDCA. The structural identification workload subsequently turned to assigning the sulfate-conjugated site, namely C3-OH and C7-OH. Fortunately, UDCA-3-sulfate was incubated with recombinase SULT2A1 in our laboratory. **M19** was justified as UDCA-3-sulfate with a reference compound. Afterward, *m*/*z* 469.3 > 97 was subjected to the first ER-MS assays, and the OCE of **M20** was determined as −101.1 eV, located within the 7-S window of the OCE distribution diagram (Figure 4C). Moreover, the backbone to the second ER-MS was matched well with UDCA. As a result, the putative identity of **M20** was assigned to UDCA-7-sulfate by applying ER^2^-MS. Additionally, **M18** was identified as 7*β*-hydroxy-3-oxo-5*β*-cholan-24-oic acid-7-suflate via the aforementioned workflow.

**M12** and **M13** were acquired as isomers through predefined ion transitions as *m*/*z* 567.3 > 391.3 at 8.83 min and 9.16 min. For one of the metabolites (**M13)**, its primary mass fragment ion species of [M–H]^−^ (*m*/*z* 567.3202) was observed at *m*/*z* 391.2863, 175.0256, 129.0200, and 113.0250 (Appendix A). After converting the *m*/*z* values to elemental compositions, **M12** and **M13** were regarded as glucuronidated products of UDCA or CDCA. Then, the structural identification workload subsequently turned to assigning the glucuronide-conjugated site, namely C3-OH, C7-OH, or C24-COOH. Afterward, the OCEs of the diagnostic ion at *m*/*z* 391.3 were calculated to be −49.96 eV and −51.50 eV for **M12** and **M13** for the first breakdown graph. Then, we observed the OCEs of two isomers located exactly within the OCE window 7-G and 3-G (Figure 4A). In depth, the scaffolds of **M12** and **M13** were confirmatively confirmed as UDCA (Figure 4D). Hence, the structures of **M12** and **M13** were identified as UDCA-7-*O*-glucuronide and UDCA-3-*O*-glucuronide through application of ER^2^-MS with UDCA.

After annotating the structures of all metabolites (**M1**–**M20**), metabolic pathways were proposed to link all metabolites (Figure 5). Overall, hydroxylation, oxidation, epimerization, sulfation, and glucuronidation were detected as the dominant metabolic pathways for the sequential metabolism of UDCA in vitro. After undergoing hydroxylation, UDCA produced two metabolites (**M7** and **M8**). The oxidation metabolism favored the C3-OH and C7-OH groups producing **M1** and **M2**. Meanwhile, two oxidation metabolites, **M4** and **M5**, resulted directly from the hydroxylation metabolites of **M7** and **M8**. Compared to C3-OH, a greater tendency was shown by C7-OH to receive epimerization, generating **M3**. **M7** was epimerized to **M6** at the C3-OH site. In addition, three sulfated metabolites (**M18**, **M19**, and **M20**) were identified, which matched well with **M1** and UDCA. Of note, **M19** was justified as UDCA 3-sulfate, which was also confirmed by the incubation of UDCA with human SULT2A1. Moreover, nine glucuronidated metabolites were identified as **M9**–**M17**. Among them, **M9** and **M10** served as the glucuronidated metabolites of **M1**. **M11** corresponded to the glucuronidated metabolites of **M2**. **M12** and **M13** were the glucuronidated metabolites of UDCA. **M14** and **M16** served as the glucuronidated metabolites of **M7**. **M15** corresponded to the glucuronidated metabolites of **M8**. **M17** was the glucuronidated metabolite of **M6**.

As we all know, bile acids are endogenous substances, which are responsible for maintaining the bile acid pool homeostasis. There is no doubt that liver microsomes could retain a little bile acid, especially high contents, such as CA and DCA, more or less, although it has been reported that the composition and hydrophobicity of the bile acid pool in mice are significantly different from those of humans [26]. The HLM and MLM incubation results shown in Appendix A demonstrate the presence of bile acids. However, there was no residue of phase II metabolites for glucuronidation or sulfation displayed in Appendix A, indicating a low level of conjugated bile acids in normal conditions. To confirm the products of UDCA sequential metabolism, UDCA-*d*_4_ was incubated in parallel. As expected, biotransformed metabolites labeled with deuterium were observed in Appendix A, except for **M19**, compared to UDCA. Hitherto, great efforts have been widely made to develop methods allowing isomeric identification, from sample preparation to detection, using cutting-edge MS/MS techniques, such as ion mobility spectrometry, chemical derivatization, and energy-resolved MS [27,28]. Although a research model using in vitro cofactor-fortified liver S9 fraction was developed to compare the metabolites of CA, CDCA, LCA, and UDCA, the bile acid conjugated sites of glucuronic acid or sulfate could not be confidently confirmed [29]. Therefore, we established an approach for isomeric structural annotation in phase II metabolites based on ER^2^-MS, which ultimately uncovered the real identities of twelve metabolites. According to a report, CYP3A4 is the primary CYP450 enzyme responsible for metabolizing both CA and CDCA in HLM [30]. To acquire a better understanding of the metabolic network within the liver, it would be helpful to assess the related isoform of UGTs or SULTs in UDCA hepatic metabolism processes. Moreover, our findings suggest that an in vitro incubation system can serve as a cost-effective alternative for obtaining bile acid glucuronide standards, which are otherwise expensive on the market.

In the current work, we developed an integrated approach via combining high-resolution MS and the *p*MRM strategy of Qtrap-MS, which facilitated the comprehensive profiling and characterization of prototype compounds and metabolites of UDCA. In addition, ER^2^-MS offers a unique dimension to aid in the annotation of isomeric structures. The results showed that the biotransformation pathways of UDCA mainly included oxidation, hydrogenation, glucuronidation, and sulfonation.

## 3. Materials and Methods

### 3.1. Materials and Chemicals

Homogenized liver microsomes of humans and mice were purchased from the Research Institute for Liver Diseases Co., Ltd. (Shanghai, China). As an internal standard (IS), astragaloside IV was obtained from Yuanye Biotech Co., Ltd. (Shanghai, China). Authentic compounds, such as 7-ketolithocholic acid, UDCA, chenodeoxycholic acid (CDCA), ursocholic acid (UCA), *β*-muricholic acid (*β*-MCA), and UDCA-*d*_4_, were commercially supplied by Yuanye Biotech Co., Ltd. (Shanghai, China). UDCA-3-sulfate (UDCA-3-S) was previously biosynthesized by SULT2A1 in our laboratory.

LC–MS grade acetonitrile (ACN), methanol, ammonium formate, as well as formic acid were obtained from Thermo-Fisher (Pittsburgh, PA, USA). Ultrapure deionized water was prepared using an in-house Millipore Milli-Q Integral water purification system (Bedford, MA, USA). Nicotinamide adenine dinucleotide phosphate (NADPH), glucose 6-phosphate (G6P), glucose-6-phosphate dehydrogenase (G6PD), uridine 5′-diphosphate glucuronic acid (UDPGA), alamethicin, dithiothreitol (DTT), D-saccharic acid-1,4-lactone monohydrate, 3′-phosphoadenosine-5′-phosphosulfate (PAPS), magnesium chloride, tris hydrochloride (Tris HCl), phosphate-buffered saline (PBS) were purchased from Yuanye Biotech Co., Ltd. (Shanghai, China).

### 3.2. Liver Microsomes Incubation Condition

Liver microsomes from humans (HLM) and mice (MLM) with a protein concentration of 20 mg/mL were thawed cautiously on ice and divided into aliquots for further experiments. In order to illustrate the biotransformation process of UDCA in vivo, this study performed sequential metabolism of UDCA in two steps. Firstly, for phase I metabolism, the NADPH regeneration system was composed of magnesium chloride (100 mM), Tris HCl (50 mM), G6P (10 mM), G6PD (50 U/mL), and NADPH (40 mM). For phase II metabolism, two-phase reactions were mainly involved in conjugating the hydroxyl groups with sulfo and glucuronyl groups individually. One UDGPA-related incubation system contained Tris HCl (50 mM), magnesium chloride (100 mM), alamethicin (5 mg/mL), D-saccharic acid 1,4-lactone (20 mM), and UDPGA (40 mM). Another PAPS-oriented incubation system consisted of PBS (pH 7.4, 50 mM), DTT (5 mM), magnesium chloride (8 mM), and PAPS (10 mM). Then, authentic BAs including UDCA at a concentration of 10 mM and UDCA-*d*_4_ labeled standard of 10 mM were added to three regeneration systems along with 0.05 mg of hepatic microsomal protein from either human or mouse, respectively. After a preincubation period of 5 min at 37 °C, the incubation reactions were initiated by adding, respectively, the NADPH, PAPS, or UDPGA and allowed to proceed at 37 °C for 2 h. The reactions were terminated with 100 μL ice ACN containing astragaloside IV (1 μg/mL) at a final volume of 200 μL. For the control group, there were no substrates or microsomes present, which could be compared to the incubation systems involving UDCA or UDCA-*d*_4_.

### 3.3. Qualitative Characterization

#### 3.3.1. Qualitative Characterization for LC–MS/MS

The incubational samples’ analysis was performed on LC–Q-Exactive-MS (Thermo-Fisher Scientific, Waltham, MA, USA). Chromatographic separations were conducted on a Waters HSS T3 Column (2.1 × 100 mm, 1.8 μm, Milford, MA, USA). The mobile phase consisting of 1 mM aqueous ammonium formate containing 0.1% formic acid (A) and ACN (B) was programmed in gradients, as follows: 0–10 min, 10–95% B; 10–10.1 min, 95–10% B; 15 min, 10% B; and flow rate, 0.2 mL/min. The injection volume was 4 μL, the column oven was set at 40 °C, and the autosampler chamber was maintained at 4 °C. MS^1^ full scan was operated at 70,000 FWHM resolution among *m*/*z* 50–750, and automatically triggered MS/MS acquisition was conducted at 17 500 FWHM resolution within *m*/*z* 50–750 through a top-10 data-dependent acquisition algorithm. The isolation width of the precursor ion was 1.5 Da. The collision energy (CE) was set at 30 eV, with a collision energy spread (CES) of 15 eV. Data acquisition was carried out using Xcalibur software (Version 2.1.0, Thermo-Fisher Scientific).

#### 3.3.2. Isomeric Identification through ER^2^-MS Strategy

First ER-MS assay. The first ER-MS was programmed to acquire first-generation breakdown graphs for the predominant MS^2^ spectral signals of each compound using a previously described method [25]. For instance, in the case of **M15**, a UDCA phase II metabolite, the significant MS^1^ spectral signal was *m*/*z* 583.31 ([M–H]^−^), with primary MS^2^ fragment signals occurring at *m*/*z* 407.28 [M–H–C_6_H_8_O_6_]^−^, 175.03 [M–H–C_24_H_40_O_5_]^−^, and 113.03 [M–H–C_24_H_40_O_5_–CH_2_O_3_]^−^. Thereafter, three ion transitions, including *m*/*z* 583.31 > 407.28, *m*/*z* 583.31 > 175.03, and *m*/*z* 583.31 > 113.03, were programmed for the first ER-MS analysis. Subsequently, a series of pseudo-ion transitions (PITs) were derived for each ion transition. Three panel PITs were generated as *m*/*z* 583.3 > 407.301, 583.3 > 407.302, 583.3 > 407.303, etc.; *m*/*z* 583.3 > 175.031, 583.3 > 175.032, 583.3 > 175.033, etc.; *m*/*z* 583.3 > 113.031, 583.3 > 113.032, 583.3 > 113.033, etc. Each PITs set was accordingly assigned progressive CEs with a step size of 2 eV among the range of −5 ~ −61eV, such as −5 eV, −7eV, and so forth. The MRM parameters are listed in Appendix A. The aligned PITs and CEs sets were imported into the monitoring list of MRM. Afterward, the MRM responses of all PITs were normalized within the ion transition candidates and then input into GraphPad Prism 8.0 software (San Diego, CA, USA) to plot the breakdown graph for each ion transition via Gaussian fitting. The OCE corresponded to the acme of the breakdown graph.

Second ER-MS measurement. The second ER-MS assay was conducted using the MRM^3^ program for annotating the structures of glucuronyl-conjugated BAs in this research. Fixed CE values were assigned on the Q2 cell, while progressive AF2 values were implemented for the LIT chamber, which produced second-generation breakdown graphs through programming a sequence of MS^3^ experiments after constructing Q1 > Q3 > Q_LIT_ ion transitions. As an illustration, **M14**, **M15**, **M16**, and **M17** were chosen as representatives of isomeric-free BAs. For either Q1 or Q3, *m*/*z* 407.3 was defined with a fixed CE of −20 eV for the Q2 cell. Fourteen MRM^3^ experiments were performed with progressive AF2 values (step size of 0.005 V), ramping up from 0.025 to 0.11 V, and *m*/*z* 407.3 with a 2 Da width was applied for each experiment. The other parameter settings followed those described in the literature [25]. The peaks of each analyte harvested from different AF2 levels were merged, normalized, and used to draw Gaussian-shaped breakdown graphs for sub-product ions with GraphPad Prism 8.0 software (San Diego, CA, USA) following the MRM^3^ assays.

The predictive multiple-reaction monitoring (*p*MRM) scan mode was utilized to screen for a wide range of BAs, which was accomplished by using a monitoring list previously constructed based on LC–Qtrap-MS. [M–H]^−^ > [M–H]^−^ ion species were selected for hydroxylation, oxidation, and epimerization products. The precursor ion transitions of glucuronide-conjugated BAs were set to [M–H]^−^ with the corresponding Q3 ions set at a neutral loss of glucuronyl residue ([M–H–176.02]^−^), 175.02 Da, and 113.02 Da, whereas [M–H]^−^ > [M–H–sulfonyl residue]^−^ and [M–H]^−^ > *m*/*z* 96.96 Da (HSO_4_^−^) were set to capture the sulfation products. Ultimately, the *p*MRM program was configured to perform qualitative analysis simultaneously involving all sequential metabolites. The LC elution program was the same as high-resolution MS.

## 4. Conclusions

Although UDCA serves as the primary drug for cholestasis treatment, its metabolism to other bile acid species has not been fully revealed using LC–MS/MS due to its inability to address the blind drawback of isomeric metabolites. To overcome this limitation, we proposed a new strategy named ER^2^-MS to structurally characterize the conjugated metabolites. This was achieved by applying well-defined mass cracking rules and authentic compounds for phase I metabolites. In terms of the given phase II metabolites, the conjugation sites were correlated with first-generation breakdown graphs corresponding to the linkage fission mediated by collision-induced dissociation, and the structural nuclei were identified by matching second-generation breakdown graphs with the known structures. In this study, we used ER^2^-MS to detect and identify twenty metabolites (**M1**–**M20**), including eight phase I metabolites and twelve phase II metabolites, after in vitro incubation of UDCA under different regeneration systems. Furthermore, the glucuronidation and sulfation conjugation sites of UDCA were confirmed via ER-MS, while the structural nuclei were identified by ER^2^-MS. More importantly, the current study can provide a legitimate analytical approach for the characterization of metabolites, particularly phase II metabolites.

## Figures and Tables

**Figure 1 molecules-28-04801-f001:**
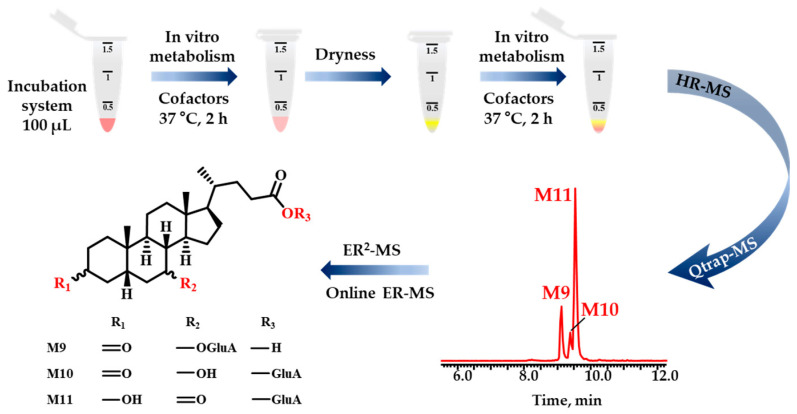
The workflow comprising three primary steps was proposed. Firstly, UDCA underwent sequential metabolism in vitro in a NADPH incubation system. The phase I metabolic products served as substrates for phase II metabolism after being evaporated to dryness with nitrogen. Secondly, the structural characterization of metabolites, even the minor ones, was accomplished by combining HR-MS with Qtrap-MS. Thirdly, the ER^2^-MS strategy was employed for structural identification of those isomers.

**Figure 2 molecules-28-04801-f002:**
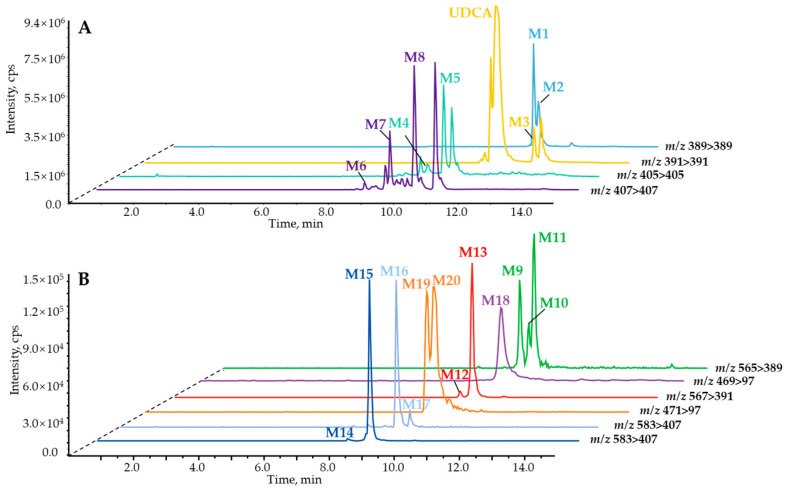
Extracted ion current chromatogram of phase I metabolites (**A**), phase II glucuronidated and sulfated metabolites (**B**) of UDCA from LC−*p*MRM program.

**Figure 3 molecules-28-04801-f003:**
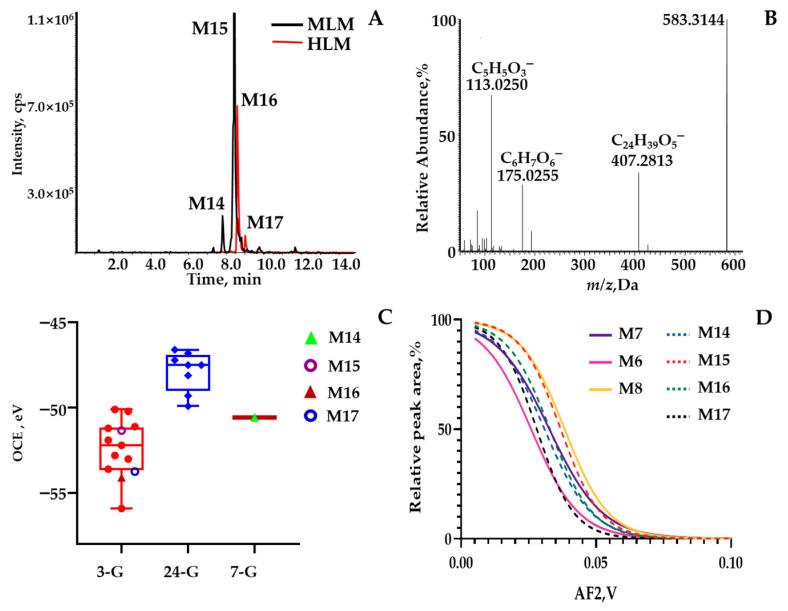
(**A**) Extracted ion current chromatogram (*m*/*z* 583.3 > 407.3) from LC−*p*MRM program; (**B**) MS^2^ spectrum of the quasimolecular ion ([M − H]^−^) of **M15** acquired by LC−MS; (**C**) Box plot for the OCE scattering pattern of *m*/*z* 583.3 > 407.3 against different glucuronidation sites (C-3, C-7, or C-24) via assaying literature in our group; (**D**) Overlaid sigmoid-shaped second-generation breakdown graphs (relative intensity of 50% against AF2 values) acquired by programming *m*/*z* 583.3 > 407.3 > 407.3 for **M6**–**M8** and **M14**–**M17**.

**Figure 4 molecules-28-04801-f004:**
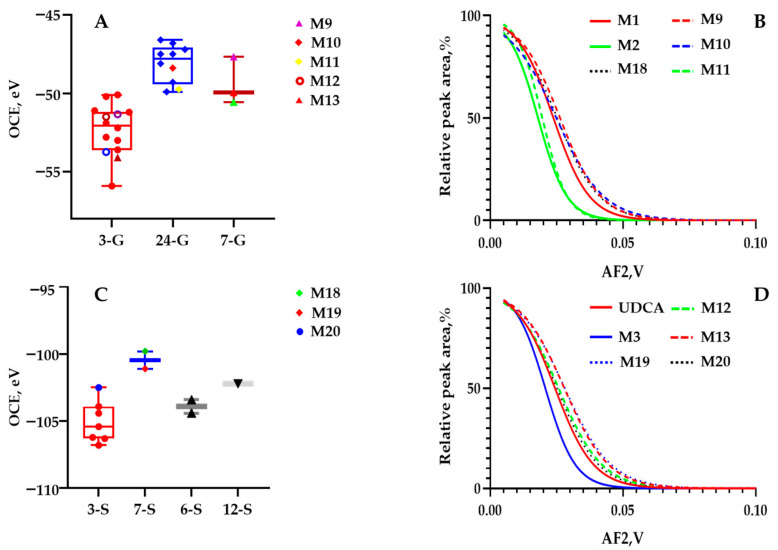
(**A**) Box plot for the OCE scattering pattern of *m*/*z* 565.3 > 389.3 and *m*/*z* 567.3 > 391.3 against different glucuronidation sites (C-3, C-7, or C-24) via assaying authentic compounds in our group; (**B**) Overlaid sigmoid-shaped second-generation breakdown graphs (relative intensity of 50% against AF2 values) acquired by programming *m*/*z* 389.3 > 389.3 > 389.3 for **M1** (red curve; equation: *Y* = 100/(1 + 10^^((0.02379−X) × −64.11))^), R^2^ = 0.9862), **M2** (green curve; equation: *Y* = 100/(1 + 10^^((0.01802−X) × −79.94))^), R^2^ = 0.9865), *m*/*z* 565.3 > 389.3 > 389.3 for **M9** (red dotted curve; equation: *Y* = 100/(1 + 10^^((0.02630−X) × −56.85))^), R^2^ = 0.9864), **M10** (blue dotted curve; equation: *Y* = 100/(1 + 10^^((0.02491−X) × −48.96))^), R^2^ = 0.9875), **M11** (green dotted curve; equation: *Y* = 100/(1 + 10^^((0.01962−X) × −92.44))^), R^2^ = 0.9672), *m*/*z* 469.3 > 97 > 97 for **M18** (black dotted curve; equation: *Y* = 100/(1 + 10^^((0.01962−X) × −92.44))^), R^2^ = 0.9672); (**C**) Box plot for the OCE scattering pattern of *m*/*z* 469.3 > 97 and *m*/*z* 471.3 > 97 against different sulfated sites (C-3, C-6, C-7, or C-12) via assaying literature in our group; (**D**) Overlaid sigmoid-shaped second-generation breakdown graphs (relative intensity of 50% against AF2 values) acquired by programming *m*/*z* 391.3 > 391.3 > 391.3 for **M3** (blue curve; equation: *Y* = 100/(1 + 10^^((0.02077−X) × −76.97))^), R^2^ = 0.9630), *m*/*z* 571.3 > 391.3 > 391.3 for **M12** (green dotted curve; equation: *Y* = 100/(1 + 10^^((0.02576−X) × −53.30))^), R^2^ = 0.9764), **M13** (red dotted curve; equation: Y = 100/(1 + 10^^((0.02794−X) × −52.25))^), R^2^ = 0.9892), *m*/*z* 471.3 > 97 > 97 for M19 (blue dotted curve; equation: Y = 100/(1 + 10^^((0.02812−X) × −49.78))^), R^2^ = 0.9951), and **M20** (black dotted curve; equation: Y = 100/(1 + 10^^((0.02516−X) × −55.55))^), R^2^ = 0.9391).

**Figure 5 molecules-28-04801-f005:**
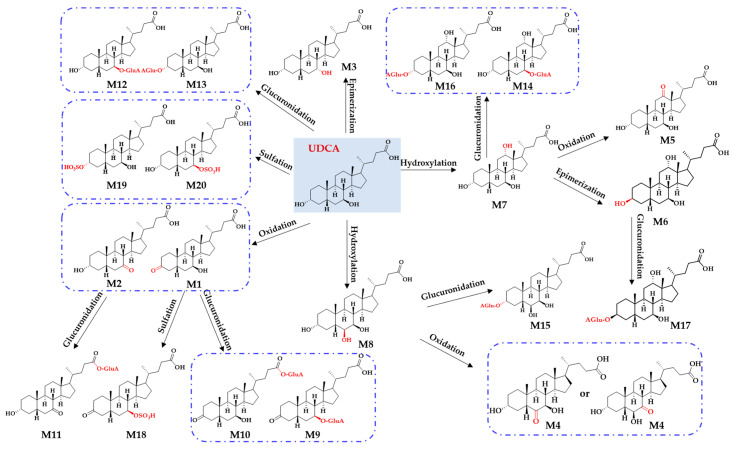
Proposed metabolic pathways, such as hydroxylation, oxidation, epimerization, sulfation, and glucuronidation, in response to the generation of twenty metabolites (**M1**−**M20**) for UDCA sequential metabolism in vitro.

## Data Availability

Not applicable.

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
