# Peer review of "Characterization of Metabolic Correlations of Ursodeoxycholic Acid with Other Bile Acid Species through In Vitro Sequential Metabolism and Isomer-Focused Identification"

_molecules, 2023, doi:10.3390/molecules28124801_

Round 1
Reviewer 1 Report
The topic of this manuscript is interesting and fits well the scope of Molecules. The reviewer feels it can be accepted after some minor amendments.
(1) Full name of UCDA should be indicated in the title.
(2) Change the 1st metabolism to metabolic in the title.
(3)Did the authors detect any species difference in UCDA metabolism between humans and mice?
(4) Could the authors quantify the major metabolite(s)?
Reviewer 2 Report
The paper is interesting, well written, well structured in all parties. Methodology is adequate and coerent with the endppints. Results and discussion are well defined. The paper is acceptable for publication in the current form
Minor english editing
Reviewer 3 Report
Observations:
-Please edit the abstract: avoid the use of acronyms or include your description.
- Include a general figure (scheme) of the experimental methodology, so that the reader can easily follow the sequence of experiments and assays carried out. Explain briefly in the text.
